# Cannabidiol Protects the Neonatal Mouse Heart from Hyperoxia-Induced Injury

**DOI:** 10.3390/ijms27010146

**Published:** 2025-12-23

**Authors:** Teresa Hellberg, Thomas Schmitz, Christoph Bührer, Stefanie Endesfelder

**Affiliations:** Department of Neonatology, Charité—Universitätsmedizin Berlin, Augustenburger Platz 1, 13353 Berlin, Germany; teresa.hellberg@charite.de (T.H.); thomas.schmitz@charite.de (T.S.); christoph.buehrer@charite.de (C.B.)

**Keywords:** neonatal hyperoxia, cardiac remodeling, cannabidiol (CBD), oxidative stress and inflammation, cardiomyocyte proliferation

## Abstract

Neonatal hyperoxia induces oxidative and inflammatory stress that disrupts cardiac maturation and contributes to long-term cardiovascular morbidity in individuals born preterm. Cannabidiol (CBD), a non-psychoactive phytocannabinoid with antioxidant and anti-inflammatory properties, has demonstrated protective effects in neonatal hyperoxic injury in other organs; however, its impact on the developing heart remains unclear. This study investigated whether CBD mitigates hyperoxia-induced cardiac injury in a neonatal mouse model. Newborn mice were exposed to 80% O_2_ for 48 h from postnatal day (P)5 to P7 and received vehicle, 10 mg/kg CBD, or 30 mg/kg CBD intraperitoneally, while controls remained in room air. Hearts were collected at P7 or after recovery until P14. Hyperoxia triggered oxidative stress (*Nrf2*), inflammation (*IL1β*, *TNFα*, *IL6*, *CXCL1*; *p* < 0.05), and dysregulated apoptosis/autophagy, leading to reduced cardiomyocyte proliferation (Ki67^+^ −50% at P14; *p* < 0.01) and adverse remodeling (hypertrophy, fibrosis; *p* < 0.01). CBD attenuated these responses and normalized autophagy (*Atg5*, *Atg12*; *p* < 0.05). Notably, 10 mg/kg CBD, but not 30 mg/kg, preserved proliferative capacity and reduced wall thickness, suggesting a narrow therapeutic window, while both doses limited collagen deposition and apoptosis (*Casp3*, *AIF*; *p* < 0.05). Several effects were sex-dependent, with males exhibiting more pronounced long-term structural and proliferative impairments and greater responsiveness to low-dose CBD. These findings identify CBD as a potential cardioprotective modulator of neonatal hyperoxia-induced injury and highlight the importance of dose- and sex-specific mechanisms in early cardiac maturation.

## 1. Introduction

Oxygen therapy remains a cornerstone of neonatal intensive care, providing life-saving support for preterm infants suffering from respiratory distress and hypoxemia [1]. However, prolonged or excessive oxygen exposure, termed neonatal hyperoxia, induces significant oxidative stress and tissue injury extending beyond the pulmonary system [2,3].

Emerging evidence indicates that the neonatal heart is particularly susceptible to oxygen toxicity, resulting in long-term alterations in cardiac structure and function that persist into adulthood [4]. Consequently, adults born preterm show a markedly elevated risk of developing heart failure and ischemic heart disease, likely due to permanent alterations in myocardial and vascular architecture [5,6]. Specifically, imaging studies reveal smaller ventricular chambers, increased myocardial mass, diastolic dysfunction, and enhanced fibrosis, providing a structural basis for this reduced cardiac reserve [7,8,9,10,11].

Rodent models of postnatal hyperoxia demonstrate that early-life exposure to high oxygen concentrations disrupts cardiomyocyte proliferation [4,12], provokes oxidative and inflammatory stress responses [13], and triggers maladaptive remodeling processes characterized by hypertrophy, fibrosis, and contractile dysfunction [14,15,16]. Collectively, these changes produce a phenotype resembling developmental cardiomyopathy, emphasizing the urgent need for therapeutic strategies that protect the developing myocardium from oxygen-induced injury [14,15,16,17,18].

The pathophysiological mechanisms underlying hyperoxia-induced cardiac damage are complex and multifactorial, with oxidative stress recognized as the central initiating event [19,20,21]. Excessive production of reactive oxygen species (ROS) overwhelms immature antioxidant defenses, thus leading to oxidative injury of lipids, proteins, and DNA [22]. Hyperoxia increases ROS generation primarily through mitochondrial and NADPH oxidase pathways, resulting in cellular dysfunction and death [2,23]. In premature infants, the enzymatic antioxidant systems are underdeveloped, rendering these patients especially vulnerable to oxidative injury [24,25]. When this defense system becomes dysregulated, either overactivated or functionally exhausted, ROS accumulation leads to sustained oxidative stress. This, in turn, not only causes damage to macromolecules but also activates inflammatory and cell death signaling cascades [26,27].

Hyperoxia-induced oxidative stress triggers a strong secondary inflammatory response, characterized by the release of pro-inflammatory cytokines and chemokines. While transient inflammation may promote repair, chronic activation drives fibrosis and interferes with postnatal cardiac maturation [15,28]. These oxidative and inflammatory cues converge on cell fate pathways: hyperoxia disrupts the balance between apoptosis and protective autophagy in the neonatal heart, promoting cardiomyocyte loss and diminishing regenerative potential [12,29]. In parallel, activation of fibroblast-associated pathways induces interstitial fibrosis and ventricular stiffening [30]. Critically, hyperoxia also suppresses the early postnatal proliferative capacity of cardiomyocytes by downregulating proliferative signaling and enhancing Hippo-mediated growth inhibition, thereby limiting cardiac growth and regeneration [12,30,31].

Given the multifaceted nature of oxygen-induced injury and the absence of targeted cardioprotective therapies, agents with broad cytoprotective potential are of particular interest [32]. Cannabidiol (CBD), a non-psychoactive phytocannabinoid derived from *Cannabis sativa*, has attracted attention for its anti-oxidative, anti-inflammatory, and anti-apoptotic properties in diverse models of tissue injury. CBD exerts these effects largely independently of the canonical cannabinoid receptors CB1 and CB2 [33], instead modulating multiple molecular targets, including peroxisome proliferator-activated receptor γ (PPARγ) [34], transient receptor potential (TRP) channels [33], and adenosine signaling pathways [35]. CBD is licensed as a medical drug for use in children with epileptic seizures [36]. Importantly, CBD enhances endogenous antioxidant defenses, suppresses inflammation, and stabilizes mitochondrial function under conditions of oxidative stress [37]. Preclinical studies have demonstrated the protective effects of CBD in models of neonatal hyperoxia-induced injury in the lung, brain, and retina, suggesting a broader potential for mitigating oxygen-induced damage in other developing organs [38,39,40]. Moreover, data from experimental animals suggest that cannabidiol might serve as an adjunct to therapeutic hypothermia in newborn infants with post-asphyxia encephalopathy and reduce rates of intraventricular hemorrhage in preterm infants [41,42].

Despite these promising findings, the role of CBD in preventing or attenuating hyperoxia-induced cardiac injury remains largely unexplored. The developing heart presents unique challenges due to its rapid cellular turnover, dynamic metabolic demands, and immature redox and inflammatory control systems [11,43,44,45]. Moreover, the dose-dependent and temporal characteristics of CBD’s effects in the neonatal context are not well defined. High doses may paradoxically suppress essential physiological stress responses or interfere with developmental signaling pathways [46,47,48,49]. Therefore, determining the optimal dosing and mechanistic specificity of CBD in the setting of neonatal oxygen injury is critical for future translational applications.

In light of these considerations, the present study aimed to comprehensively delineate the molecular, structural, and functional consequences of neonatal hyperoxia exposure in the developing mouse heart, and to determine whether CBD can mitigate these effects in a dose- and time-dependent manner. To our knowledge, this is the first study to systematically evaluate the cardioprotective potential of CBD in the context of oxygen-induced cardiac injury during early postnatal development.

## 2. Results

### 2.1. Regulation of Oxidative Stress

We observed a significant upregulation of the anti-oxidative defense system on postnatal day (P) 7. The expression of *Nrf2* (Figure 1A), the master transcription factor regulating the cellular defense against oxidative stress, was significantly increased upon hyperoxia exposure. CBD at both applied concentrations (10 mg/kg and 30 mg/kg) was able to reduce this *Nrf2* induction, thereby mitigating the cellular indicator of oxidative stress. On P14, the hyperoxia-induced stress response was no longer detectable for *Nrf2*. No effects were observed on *Nrf2* expression following hyperoxia exposure or CBD treatment (Figure 1A).

In the downstream target of *Nrf2*, *Hmox1*, the inducible isoform responsible for catabolizing pro-oxidative heme into the anti-oxidative molecule bilirubin, showed an inhibition of transcription following 80% oxygen exposure on P7 (Figure 1B). CBD at the 30 mg/kg concentration was capable of reversing this inhibition of *Hmox1* transcription. No effect of hyperoxia was detectable for *Hmox1* at P14. However, in this later phase, 10 mg/kg CBD treatment induced *Hmox1* expression compared to the hyperoxia-only group, indicating a potentially different, delayed regulatory mechanism of CBD on *Hmox1* that is independent of the acute hyperoxia-induced inhibition (Figure 1B).

### 2.2. Impact on Inflammatory Cytokine and Chemokine Expression

Hyperoxia significantly altered the expression of several inflammatory mediators on P7 after 2 days of hyperoxia exposure. *IL1β*, a crucial pro-inflammatory cytokine that mediates the inflammatory response and often increases with oxidative stress, had its mRNA expression elevated following hyperoxia exposure (Figure 2A). This effect was successfully reduced by CBD at a concentration of 30 mg/kg under hyperoxic conditions. Similarly, *TNFα*, a major pro-inflammatory cytokine known for its role in cell recruitment and signaling pathways linked to oxidative stress, was induced by hyperoxia (Figure 2B).

Treatment with CBD 30 mg/kg restored its expression to normoxic control levels. However, the lower concentration of CBD (10 mg/kg) did not affect hyperoxia-induced *TNFα* expression. In contrast, the hyperoxia-induced increase in *IL6* mRNA, a pleiotropic (both pro- and anti-inflammatory) cytokine often associated with the acute-phase response, was counteracted by both CBD treatments. Both concentrations successfully reduced expression back to control levels (Figure 2C). Most dramatically, the expression of the neutrophil-recruiting chemokine, *CXCL1* mRNA, was significantly induced by hyperoxia, increasing 2.5-fold compared to the expression levels observed in the normoxic controls (Figure 2D). Both CBD concentrations were effective in preventing this induction, with CBD 30 mg/kg reducing the transcription to normoxic levels.

On P14, the inflammatory response profile differed significantly (Figure 2A–D). Hyperoxia exposure alone did not increase the expression of *IL1β*. Nevertheless, CBD 30 mg/kg reduced the non-elevated baseline concentration of *IL1β* mRNA (Figure 2A). No effects were detected for either hyperoxia or CBD treatment on the expression of *TNFα*, *IL6*, or *CXCL1* on P14 (Figure 2B–D). The data suggest a diminished inflammatory transcriptomic response to both hyperoxia and CBD by P14.

### 2.3. Effects on Apoptotic and Autophagic Signaling Pathways

We further analyzed key markers of apoptosis and autophagy on P7. Caspase-3 (*Casp3*), a central effector protease that executes apoptosis, did not show increased expression under hyperoxia. However, CBD treatment at both the 10 mg/kg and 30 mg/kg concentrations reduced *Casp3* expression under hyperoxic conditions, even falling below the level of the control animals (Figure 3A). In contrast, *AIF*, a caspase-independent mediator of nuclear apoptosis that translocates from the mitochondria, was induced by hyperoxia (Figure 3B). CBD 10 mg/kg successfully reduced *AIF* transcription to normoxic levels, while the 30 mg/kg concentration decreased transcription even below normoxic levels. Regarding autophagy, both *Atg5* (Figure 3C) and *Atg12* (Figure 3D), two proteins essential for the formation and elongation of the autophagosome membrane during the cell-protective process of autophagy, showed inhibited expression following hyperoxia exposure. For *Atg5*, CBD 10 mg/kg showed a tendency to promote transcription. CBD 30 mg/kg successfully elevated expression to control levels under hyperoxia (Figure 3C). In the case of *Atg12*, CBD 30 mg/kg was found to be effective in reversing the hyperoxia-induced inhibition. In contrast, CBD 10 mg/kg could not effectively counteract this effect (Figure 3D).

On P14, a distinct long-term response was observed for the apoptosis and autophagy markers, even in the absence of sustained acute oxidative stress. *Casp3* expression was elevated by prior hyperoxia exposure, indicating a delayed or sustained pro-apoptotic signaling. CBD at both the 10 mg/kg and 30 mg/kg concentrations prevented this delayed increase (Figure 3A). For *AIF*, a paradoxical sustained decrease in expression compared to the controls was observed after hyperoxia, suggesting a long-term alteration in mitochondrial signaling. Both CBD concentrations further intensified this inhibitory effect (Figure 3C). Similarly, for *Atg12*, hyperoxia caused a sustained decrease in expression compared to the controls, and both CBD concentrations further strengthened this inhibitory effect (Figure 3C). No effects were observed for *Atg5* on P14 (Figure 3D).

### 2.4. Morphometric and Histological Parameters of Cardiac Injury

To examine the structural consequences of neonatal hyperoxia and the early effects of CBD treatment on cardiac remodeling, histological analyses were carried out through postnatal day 14 (P14). We used HE staining to evaluate tissue architecture, cellular morphology, and left ventricular geometry. In addition, Sirius Red staining was employed to specifically visualize and quantify collagen deposition (myocardial fibrosis).

Representative HE-stained sections confirmed the effect of prior hyperoxia on tissue architecture and cellular morphology (Figure 4, left panels). The normoxia control exhibited a regular cellular architecture and homogeneous tissue structure. In the hyperoxia heart, the tissue appeared slightly disorganized with evidence of isolated cellular swelling or disorganization. This is indicative of an underlying injury and hypertrophy, consistent with the quantified increase in left ventricular wall thickness (see Figure 5C). The CBD-treated hearts showed differential effects: The pup treated with CBD 10 mg/kg displayed an improved tissue organization and a visibly reduced ventricular wall thickness compared to hyperoxia, supporting the efficacy of this concentration in counteracting sustained hypertrophy. Conversely, the hyperoxia heart treated with 30 mg/kg CBD showed a denser, more compact tissue structure and sustained wall thickness comparable to hyperoxia, indicating that the higher dose did not prevent this persistent form of remodeling.

Sirius Red staining confirmed the persistent fibrotic remodeling on P14. Compared to the fine, evenly distributed collagen fibers in the normoxia control, the hyperoxia heart tissue showed visibly intensified red staining and a marked increase in dense collagen fibers, supporting the quantitative data on collagen area (see Figure 5D). In the CBD-treated heart tissues, the visual findings were complex. Although quantitative analysis (Figure 5D) suggested both CBD concentrations effectively counteracted the hyperoxia-induced fibrotic area (Figure 4, right panels). To further investigate the molecular signaling driving this remodeling, we analyzed the expression of *TGFβ*. While hyperoxia induced a significant upregulation of *TGFβ* transcripts on P7, an effect that was successfully suppressed by CBD treatment (see Appendix A), no significant differences in *TGFβ* mRNA levels were detectable by P14 (Appendix A).

Measurements taken immediately after the 48-h hyperoxia exposure showed acute changes in cardiac mass. Hyperoxia significantly reduced the cardiac wet weight (heart weight after excision) on P7 compared to normoxic controls (Figure 5A). Treatment with CBD at both concentrations (10 mg/kg and 30 mg/kg) prevented this reduction, restoring the heart weights to normoxic control levels. The heart-to-body weight ratio showed a similar trend. Hyperoxia reduced the ratio on P7. However, only CBD 30 mg/kg effectively prevented this impairment, resulting in a ratio comparable to that of the control animals. CBD 10 mg/kg did not influence the impaired ratio, which remained significantly lower compared to the control group.

Histological analysis of the left ventricular myocardium and Sirius Red staining on P14 provided a visual correlation with the sustained morphometric findings (see Figure 4). Analysis of the left ventricular wall thickness on P14 (Figure 5C) revealed sustained effects of the prior hyperoxia exposure, even after recovery in room air. Hyperoxia increased the thickness of the left ventricular wall, consistent with the representative images quantified from the HE-stained tissue sections. CBD 10 mg/kg successfully counteracted this hypertrophy, whereas CBD 30 mg/kg did not influence this persistent effect.

Histological quantification of myocardial fibrosis, represented by the Sirius Red-stained area, demonstrated a clear, lasting injury. Hyperoxia impressively induced collagen deposition at P14 compared to normoxic levels. Both CBD concentrations effectively counteracted this effect. The quantification of the stained area strongly corroborates the findings visible in the representative tissue sections (see Figure 4). In contrast, on P14, no sustained effects were detected for either heart weight after excision or the heart-to-body weight ratio in any group, neither hyperoxia nor CBD.

### 2.5. Gene Expression Changes in Cardiac Remodeling and Maturation

The expression of *Col1a1*, a primary component of the extracellular matrix and a marker for myocardial fibrosis, was strongly induced after 48 h of hyperoxia at P7 (Figure 6A). Unexpectedly, treatment with both CBD 10 mg/kg and CBD 30 mg/kg not only failed to effectively counteract this induction but resulted in an “exhausted” reduction in transcription below control levels. Conversely, *CTGF*, which promotes fibrosis and angiogenesis, was reduced by hyperoxia. Neither CBD concentration influenced this marker, and its expression remained strongly reduced compared to the controls (Figure 6B).

*HIMF*, a potent inducer of angiogenesis and vasoconstriction, showed a drastic induction of transcripts following hyperoxia (Figure 6C). While CBD at both concentrations reduced this induction, the expression levels remained strongly elevated compared to the control group. Hyperoxia exposure induced the expression of both *Myh6* and *Myh7* (Figure 6D,E), which are key components of the sarcomere and markers for myocardial contractility and maturation shift. *Myh6* expression was induced by hyperoxia, and both CBD treatments counteracted this increase, with CBD 10 mg/kg reducing expression to normoxic levels and CBD 30 mg/kg reducing it below normoxic levels (Figure 6D). *Myh7* was also induced by hyperoxia, but here, only CBD 30 mg/kg was able to prevent this elevation (Figure 6E). When examining the *Myh6/Myh7* ratio (Figure 6F), which reflects the shift from the neonatal to the mature cardiomyocyte phenotype, hyperoxia had no discernible effect on the ratio itself. However, both CBD concentrations reduced this ratio below control levels.

A sustained fibrotic response was evident on P14, with *Col1a1* expression remaining elevated after recovery in room air. Crucially, at this later time point, both CBD concentrations effectively counteracted this persistent elevation (Figure 6A). The expression of *CTGF* returned to baseline, with no effects observed for hyperoxia or CBD treatment on P14 (Figure 6B). For *HIMF*, expression appeared exhausted on P14 following hyperoxia, and CBD 10 mg/kg did not facilitate recovery. Interestingly, CBD 30 mg/kg demonstrated an inducing effect at this stage, preventing the complete transcriptional exhaustion (Figure 6C). *Myh6* expression remained persistently induced by prior hyperoxia exposure on P14. Both CBD concentrations were effective in reducing this sustained transcription, keeping the expression levels below the control level (Figure 6D). *Myh7* expression showed no sustained effects on P14 (Figure 6E). Finally, the *Myh6/Myh7* ratio was persistently elevated on P14 under hyperoxia (Figure 6F). Both CBD concentrations diminished this effect, with CBD 10 mg/kg reducing the ratio to normal levels and CBD 30 mg/kg reducing it below the control level.

### 2.6. Effects on Cardiomyocyte Proliferation and Cell Cycle Regulation

The proliferative capacity of cardiomyocytes in the postnatal heart is a critical determinant of cardiac growth and regenerative potential. To assess potential differences in cell proliferation among the experimental groups, Ki67 immunostaining was performed (Figure 7).

In the control group (NO, Figure 7, upper left), consistent Ki67-positive (red) nuclei were detected, indicating the expected basal proliferative activity at this postnatal stage. In the hyperoxia (HY; Figure 7, lower left) heart, the number of Ki67-positive cells appeared slightly reduced compared to the control, suggesting a decline in proliferative capacity following injury. In contrast, the HY10 tissue (10 mg/kg CBD treated; Figure 7, upper right) displayed a higher number of Ki67-positive cells compared to hyperoxia (HY), frequently showing locally clustered proliferating cells, which indicates a partial reactivation of cell proliferation. In the heart tissue of a 30 mg/kg CBD treated pup (HY30; Figure 7, lower right), the number of Ki67-positive cells was comparable to that observed in HY, suggesting no significant restorative effect at the higher CBD concentration.

Overall, the qualitative assessment revealed a trend of reduced proliferation in the hyperoxia-exposed mouse pups, followed by a restorative increase in the pups treated with 10 mg/kg CBD, and a comparable level of proliferative activity in the pups treated with 30 mg/kg CBD relative to the hyperoxia injury group.

Corresponding to this, the number of Ki67-positive cells in relation to the area, as well as transcripts relevant to proliferation, were quantified. Immunohistochemical quantification of Ki67 as a marker for proliferating cells in the heart sections revealed that the 48-h hyperoxia exposure reduced the proliferative capacity by half on P14, indicating a lasting impairment (Figure 8A). CBD 10 mg/kg effectively counteracted this reduction, restoring proliferation toward control levels.

On P7, hyperoxia exposure reduced the expression of key cell cycle regulators. *CycD1*, which controls the G1-to-S phase transition, was diminished by hyperoxia (Figure 8B). In a strong response, both CBD concentrations not only reversed this reduction but also induced expression that surpassed control levels. In contrast, *CycD2* expression was also reduced by hyperoxia (Figure 8C), yet neither CBD concentration influenced this effect, resulting in persistently decreased expression. In addition, CBD 30 mg/kg appeared to even further reduce *CycD2* expression. *Gas2l3*, which is involved in cytoskeleton organization and potentially cell cycle regulation, was reduced by hyperoxia (Figure 8D). Only the CBD 30 mg/kg concentration was effective in counteracting this inhibitory effect.

The hyperoxia-induced reduction in *CycD1* expression persisted until P14. However, at this later stage, only CBD 30 mg/kg was able to effectively counteract this sustained reduction (Figure 8B). The hyperoxia-induced reduction in *CycD2* also persisted on P14 (Figure 8C). Notably, the effects of CBD shifted at this later time point: both CBD 10 mg/kg and CBD 30 mg/kg effectively countered the persistent reduction, with CBD 10 mg/kg normalizing expression and CBD 30 mg/kg elevating it beyond normoxic control levels. The hyperoxia-induced reduction in *Gas2l3* expression also persisted on P14 (Figure 8D). Unlike the early response, neither CBD concentration was able to abolish this persistent reduction on P14.

### 2.7. Regulation of Cardiac Function and Growth Signaling

To comprehensively assess the molecular consequences of hyperoxia and CBD treatment, we analyzed key markers of myocardial contractility and fundamental pathways regulating organ size and cellular fate, specifically the Hippo signaling pathway.

The expression of *Tnni1*, typically the fetal/re-expressed isoform in injured or stressed adult hearts, was unaffected by hyperoxia on P7 (Figure 9A). However, both CBD concentrations inhibited its transcription under hyperoxia exposure. In contrast, *Tnni3*, the primary functional isoform in mature cardiomyocytes, showed reduced expression following hyperoxia (Figure 9B). Both CBD concentrations improved this transcription, with CBD 10 mg/kg being more effective than 30 mg/kg. Hyperoxia significantly impacted components of the Hippo signaling pathway, which controls organ size and proliferation. *Cul7*, an E3 ubiquitin ligase with roles in cell cycle and growth, showed reduced expression under hyperoxia (Figure 9C). Both CBD concentrations had a strong influence, inducing expression that surpassed control levels. *Lats2*, a central tumor suppressor and key kinase in the Hippo pathway, was induced by hyperoxia (Figure 9D). Both CBD concentrations effectively counteracted this induction, but reduced transcription below control levels.

*TEAD1*, a downstream effector of the Hippo pathway that promotes proliferation, was reduced by hyperoxia (Figure 9E). Both CBD concentrations successfully counteracted this effect, returning transcription to control levels. Finally, *YAP1*, an oncogenic transcriptional co-activator opposed by the Hippo pathway, was induced by hyperoxia (Figure 9F). Only CBD with a concentration of 30 mg/kg was able to counteract this effect, reducing the transcription to levels below control.

After recovery in room air, the hyperoxia-exposed animals showed a sustained induction of *Tnni1* transcription on P14 (Figure 9A). CBD at both concentrations was able to counteract this delayed induction. No effects were detectable for *Tnni3* on P14 (Figure 9B). The induction of *Lats2* in response to hyperoxia was observed to persist until P14. (Figure 9D). Both CBD concentrations were able to counteract this persistent effect, reducing expression back to control levels. For *YAP1*, hyperoxia alone had no detectable effect on P14 (Figure 9F). However, both CBD concentrations reduced its transcription below control levels, suggesting a persistent inhibitory effect of CBD on this pro-growth pathway. No effects were detectable for *Cul7* or *TEAD1* on P14 (Figure 9C,E).

In an examination of the *Tnni1/Tnni3* ratio (Appendix A), an essential effect during cardiac maturation and adaptation to the altered physiological demands after birth, an increase was observed under hyperoxia on P7, persisting until P14. Both CBD concentrations lowered this ratio to the control values on P14, and CBD with the higher concentration lowered the ratio below the control level on P7.

### 2.8. Sex-Specific Differences in the Cardiac Response to Hyperoxia and CBD Treatment

Sex-stratified analyses revealed pronounced and temporally dynamic differences in the transcriptional response to hyperoxia and its modulation by CBD (see Appendix A). On P7, female pups showed a stronger sensitivity to oxidative stress-related signaling pathways, with hyperoxia selectively inducing *Nrf2*, *CTGF*, *Myh7*, *Cul7*, and *IL6* in females, whereas males responded predominantly through *Myh6* induction. In line with this, CBD treatment elicited sex-dependent protective effects. Females displayed a broader CBD-responsive profile. Low-dose (10 mg/kg) and high-dose (30 mg/kg) CBD effectively modulated *Nrf2*, *Myh7*, *HIMF*, and *IL6*, while high-dose CBD additionally influenced *CycD2*. In contrast, males exhibited CBD responsiveness mainly for *CTGF* (high-dose) and *Myh6* (both doses), indicating that early cardioprotective mechanisms are distinctly sex-biased on P7, with females showing greater transcriptional plasticity under both injury and treatment conditions.

By P14, sex differences became even more pronounced and involved genes associated with long-term injury, remodeling, and cell cycle regulation. Hyperoxia induced sustained alterations primarily in males, including increased transcription of *AIF*, *Casp3*, *Col1a1*, *Myh6*, *Tnni1*, and *IL1β* and reduced *Gas2l3*, whereas females showed persistent changes mainly in *Atg12*, *Gas2l3*, and *Col1a1*. Functional markers mirrored these transcriptional patterns: hyperoxia-induced hypertrophy (left ventricular wall thickness) and impaired cardiomyocyte proliferation (Ki67-positive cells) were evident only in males. CBD treatment also showed sex-specific efficacy in this later phase. In females, both doses modulated *AIF*, while low-dose CBD reduced *Col1a1*, and high-dose CBD influenced neither *Col1a1* nor *IL1β*. In males, the CBD effects were more restricted. The low-dose CBD affected *Casp3*, the high-dose CBD modulated *Col1a1* and *IL1β*, and only males showed recovery of Ki67 levels in response to low-dose CBD. Collectively, these findings indicate that males exhibit a stronger and more persistent hyperoxia-induced injury profile, whereas females show earlier but more targeted responses, with CBD exerting dose- and sex-dependent regulatory effects across oxidative stress, remodeling, and proliferative pathways.

## 3. Discussion

In the present study, we demonstrate that exposure of newborn mice to supraphysiological oxygen concentrations sustained cardiac injury that persisted into juvenile age. This injury is characterized by cardiomyocyte hypertrophy, interstitial fibrosis, reduced proliferative capacity, and persistent activation of oxidative stress, inflammatory, and remodeling-associated signaling pathways. A central finding of this study is that prophylactic CBD treatment significantly attenuated these pathological changes, albeit in a dose- and pathway-dependent manner. While both CBD doses reduced markers of oxidative stress and inflammation during the acute injury phase (P7), distinct long-term effects emerged on P14. The lower dose (10 mg/kg) more effectively preserved cardiomyocyte proliferation and prevented maladaptive hypertrophic remodeling, whereas the higher dose (30 mg/kg) showed greater efficacy in suppressing acute inflammatory and autophagic inhibition but failed to fully normalize growth-related signaling. These findings suggest the existence of a critical therapeutic window for CBD in the neonatal heart, in which effective attenuation of injury must be balanced against the preservation of physiological growth and developmental signaling.

### 3.1. Mechanistic Interpretation of Oxidative Stress, Inflammation, and Cell Survival

Oxidative stress is a well-established contributor to organ damage in newborns exposed to high oxygen levels. This vulnerability is linked to immature antioxidant capacity, high mitochondrial oxygen consumption, and a relatively low abundance of radical scavengers in the neonatal myocardium [23]. In our model, hyperoxia activated redox-sensitive signaling pathways, including the transcription factor *Nrf2*, and suppressed transcription of the antioxidant gene *HMOX1*. This finding is consistent with previous work demonstrating a heightened susceptibility of neonatal rodent organs to oxidative stress [50,51,52,53,54,55,56,57,58]. CBD reversed these changes, suggesting restoration of redox homeostasis. This aligns with reports that CBD upregulates antioxidant defenses via Nrf2/ARE signaling and reduces ROS/RNS formation in neuronal and cardiovascular models. For example, CBD reduced lipid peroxidation and prevented depletion of endogenous antioxidants in a mouse model of perinatal asphyxia [59]. In cerebral ischemia, CBD decreased oxidative stress in a dose-dependent manner and mitigated cell damage [60]. Similar effects have been observed in diabetic cardiomyopathy models and human cardiomyocytes, in which CBD alleviated oxidative/nitrative stress and reduced myocardial dysfunction, inflammation, fibrosis, and cell death [61].

Inflammation is closely interconnected with oxidative stress and contributes to impaired tissue regeneration. In our model, hyperoxia induced significant upregulation of *IL1β*, *TNFα*, and *CXCL1*. CBD treatment suppressed these cytokines, particularly at the higher dose, indicating anti-inflammatory activity. In preclinical cardiovascular models, CBD has been shown to reduce macrophage activation, inhibit pro-inflammatory cytokine release, and attenuate oxidative stress and cell death signals [62].

In the aforementioned diabetic cardiomyopathy model, CBD attenuated the inflammatory response and reduced cytokine expression by modulating NF-κB and MAPK signaling [61]. In rats with LPS-induced systemic inflammation, CBD protected the heart by reducing inflammatory cell infiltration and limiting tissue injury [63]. CBD also mitigated inflammation in mice with exercise-induced myocardial damage, activating the Keap1/Nrf2/HO-1 signaling pathway [64]. Moreover, in vitro, CBD inhibited LPS-stimulated cytokine release in macrophages and prevented sodium channel dysfunction in human cardiomyocytes exposed to inflammatory mediators [65].

Taken together, the dual antioxidant and anti-inflammatory effects of CBD likely contribute to the observed mitigation of hyperoxia-induced cardiac damage. CBD also improved cell survival signals: the autophagy markers *Atg5*/*12*, suppressed by hyperoxia, were restored by CBD, and the caspase-independent apoptosis mediator *AIF* was reduced. These findings suggest that CBD helps maintain cellular homeostasis in the neonatal myocardium under oxidative stress.

### 3.2. Structural and Functional Effects of Cardiac Remodeling

A one-week recovery period following hyperoxia resulted in left ventricular cardiomyocyte hypertrophy, increased fibrosis, and reduced cardiomyocyte proliferation in juvenile mice. These structural changes reflect maladaptive cardiac remodeling in the neonatal period, potentially predisposing to later cardiomyopathy, reduced regenerative capacity, and impaired cardiac reserve. The persistence of these alterations beyond the immediate neonatal phase underscores the risk that early-life oxygen exposure programs long-term heart disease [9,11,17,66].

The regulation of troponin I isoforms provides additional insight into the functional consequences of this remodeling. The persistent or reactivated expression of *Tnni1*, which is typically restricted to the fetal phase, is indicative of cardiomyocyte dedifferentiation and ongoing pathological remodeling [67]. Accordingly, the reduction in *Tnni1* expression under CBD treatment suggests suppression of fetal stress programs and functional normalization of the myocardial phenotype [68]. Although low *Tnni3* expression is expected in the early neonatal period, persistently reduced levels beyond the physiologic maturation window reflect disrupted transcriptional and epigenetic regulation of isoform switching and may contribute to cardiac dysfunction [69,70]. The developmental switch from *Tnni1* to *Tnni3* is essential for postnatal myocardial maturation and adaptation [71]. While hyperoxia-induced *CTGF* upregulation indicates activation of stress and extracellular matrix modulation, its contribution to fibrosis appears limited in this context [72,73].

In contrast, our findings support a central role for *TGFβ* in the early phase of oxygen-induced remodeling. Hyperoxia strongly induced *TGFβ* signaling in cardiac fibroblasts via oxygen-sensitive pathways, promoting myofibroblast differentiation and extracellular matrix deposition through Smad2/3 signaling [74,75,76]. We observed a transient *TGFβ* induction on P7 that was effectively suppressed by CBD, with normalization by P14, indicating that pro-fibrotic signaling is an early, time-limited event. Despite this normalization, reduced fibrosis persisted on P14 in CBD-treated animals, suggesting that CBD acts by inhibiting an early priming phase of myofibroblast activation. This temporal dissociation highlights the importance of early intervention in neonatal oxygen injury.

Importantly, CBD at 10 mg/kg prevented hypertrophy and fibrosis and preserved proliferative capacity, indicating a favorable structural and regenerative outcome. In contrast, 30 mg/kg reduced oxidative and inflammatory markers but did not fully restore proliferation or normalize structural indices. CBD has also been shown to enhance cardiomyocyte proliferation and heart regeneration after myocardial infarction in mice at 10 mg/kg, with minimal effects at lower doses [77]. These findings suggest a therapeutic window for CBD in neonatal cardiac protection: excessively low doses may be ineffective, whereas excessively high doses may interfere with physiological growth or regenerative signaling.

### 3.3. CBD Effects on Hippo/YAP and Growth Signaling

An important aspect of our findings is the modulation of developmental signaling pathways by CBD, particularly the Hippo/YAP axis, cyclin D1, and cardiomyocyte proliferation. Hyperoxia reduced proliferation, suppressed *cyclin D1*, increased inhibitory Hippo/YAP components such as *Lats2*, and decreased *YAP1* transcription. CBD at 10 mg/kg normalized these changes and restored proliferation, as described above. Despite reduced damage markers, the 30 mg/kg dose resulted in sustained *YAP1* suppression on P14. In a study by Ren et al. [77], CBD modulated the Hippo/YAP signaling pathway after myocardial infarction by increasing YAP target genes, including *Yap* and *Ctnnd1*. Similarly, CBD mitigates doxorubicin-induced myocardial damage, subsequent oxidative stress, and apoptosis in H9c2 cells and mouse models, acting partly through Hippo signaling regulation [78].

These findings suggest that while CBD attenuates damaging signals, higher doses or administration during a critical developmental window may also suppress growth-promoting pathways. This warrants caution, as controlled hypertrophy and proliferation are necessary for normal neonatal cardiac growth, and excessive suppression may impair compensatory responses and long-term recovery.

### 3.4. Gender-Specific Differences in Response to Hyperoxia and CBD

Our data clearly indicate that sex-specific factors substantially influence the cardiac response to hyperoxia and the modulatory effects of CBD. These differences align with the established concept that male and female neonatal hearts undergo divergent maturation trajectories early in life, particularly in mitochondrial function, oxidative stress processing, inflammation regulation, and cell cycle control. These differences contribute to distinct thresholds of stress resilience and regenerative capacity [79,80,81]. Female neonatal cardiomyocytes display lower apoptosis rates and better recovery after oxidative stress, whereas male hearts show greater susceptibility to mitochondrial dysfunction and oxidative injury [80,82].

The observation that CBD affects different signaling pathways depending on sex and developmental stage suggests that therapeutic interventions in the neonatal heart are not gender-neutral. Mechanistically, this may relate to sex-dependent differences in antioxidant signaling, fibroblast activity, immune cell recruitment, and metabolic phenotype—areas that are particularly sensitive to both external stressors and pharmacological agents in early postnatal life [83,84,85].

From a translational perspective, these findings highlight the importance of considering sex-related differences in the maturation and stress susceptibility of the human preterm heart, especially in the context of oxygen exposure and potentially protective therapies. Sex-specific assessment of mechanisms, dosage windows, and long-term effects may be essential for predicting the therapeutic efficacy of CBD in neonatal care.

### 3.5. Translational Relevance and Therapeutic Potential

Our findings support the hypothesis that CBD may represent a promising adjunctive therapy to protect the neonatal heart from oxygen-induced injury. Preterm infants frequently require oxygen supplementation and are at risk of oxidative organ damage, including cardiac sequelae; thus, the translational potential of CBD is considerable. Preclinical studies also demonstrate vasoprotective, anti-remodeling, and antioxidant effects of CBD in adult cardiovascular models [62]. Clinical evidence for CBD in neonatal cardiac protection is lacking, as no trials have evaluated CBD in preterm infants. Currently, CBD (Epidiolex) is approved in the US for specific epilepsy syndromes in children aged one year and older. Nonetheless, preclinical neonatal mouse and rat studies report encouraging cardioprotective effects.

However, extrapolation to human neonates raises important questions, including optimal dosing and timing. Our data indicate a narrow therapeutic window. Moreover, long-term functional outcomes, such as echocardiography, hemodynamics, and exercise capacity, remain to be evaluated beyond early structural endpoints. Safety considerations specific to the neonatal period, including developmental, endocrine, and pharmacokinetic aspects, are critical and insufficiently understood.

### 3.6. Limitations

Several limitations of this study must be acknowledged. Although we assessed molecular and structural endpoints, hemodynamic assessment and overall cardiac function were not examined in this pilot in vivo study. This limits immediate conclusions about the physiological relevance of the observed structural improvements and the full extent of CBD’s cardioprotective potential. The observation period ended on P14, corresponding to murine juvenile age. Mouse cardiac development in the first two postnatal weeks progresses much more rapidly than in humans. On P5, when oxygen exposure begins, mouse cardiomyocytes are still highly proliferative, and the heart remains strongly regenerative, but the critical transition phase is already underway. By P7, cardiomyocyte proliferative capacity is reduced, roughly corresponding to early infancy in humans. By P14, the mouse heart has lost regenerative ability and resembles an adult mammalian heart, whereas the human heart continues proliferating for several months [86]. Despite these limitations, this pilot study provides essential data on the general effects of CBD on the developing myocardium. These findings will serve as a basis for future studies that will include detailed hemodynamic monitoring to better understand the functional outcomes and long-term consequences for former preterm infants who reach adulthood.

This study assessed only two CBD doses (10 and 30 mg/kg), consistent with the literature. Although protective or partially protective effects were observed, a broader dose–response assessment, including timing, would help define the therapeutic window more precisely. Additionally, although this study identified antioxidant, anti-inflammatory, autophagy/apoptosis, and proliferation-related pathways, the exact molecular targets of CBD in the neonatal myocardium remain incompletely understood. Recent reviews emphasize that CBD’s mechanisms of action vary by tissue, dose, and disease context.

### 3.7. Future Research Directions

Future studies should extend follow-up into adolescence and adulthood, as long-term data on CBD’s cardiac effects are limited despite promising animal and early clinical findings. Systematic dose–response and timing studies are necessary to define the therapeutic window, as no clinically established CBD dosage exists for cardiac indications. Further elucidation of the underlying mechanisms remains a key objective. Integrating these aspects is essential to validate CBD’s therapeutic potential in neonatal cardiac injury and to inform future intervention studies.

## 4. Materials and Methods

### 4.1. Animal Welfare

The standardized mice utilized for this experimental series were *C57BL/6NRj* purchased from Janvier Lab SAS (Le Genest-Saint-Isle, France). Timed-pregnant dams were individually housed upon arrival and given at least one week to acclimatize before their expected delivery date under strictly controlled environmental conditions, which included a continuous ambient temperature, a relative humidity consistently maintained at 60%, and a constant 12-h/12-h light–dark cycle.

All animals were provided with ad libitum access to standard food (Altromin Spezialfutter GmbH & Co. KG, Lage, Germany, Cat. 1324, with 24% protein, 11% fat, 65% carbohydrates) and water. Following birth, all pups were kept with their lactating mothers until the designated time points. All procedures involving live animals received explicit approval from the local animal welfare authority (Landesamt für Gesundheit und Soziales Berlin, Germany, G-0014/23). The entirety of the animal experimentation was carried out in strict accordance with the internal guidelines established by the Charité Universitätsmedizin Berlin and fully complied with the ARRIVE guidelines (Animal Research: Reporting of In Vivo Experiments), as detailed by Percie du Sert et al. [87].

### 4.2. Experimental Study Design

On postnatal day 2 (P2), the pups were randomly allocated across different litters and sexes and subsequently transferred, along with the mothers, to new cages. The assignment of the newborn mice to the two distinct oxygen exposure groups, normoxia (21% O_2_) and hyperoxia (80% O_2_), was performed according to a random principle during this initial randomization. To ensure adequate care, a maximum of six pups were assigned per lactating dam. Drug or vehicle applications were administered on P5, P6, and P7, with the corresponding oxygen exposure (normoxia or hyperoxia) commencing on P5 and continuing for a total of 48 h until P7. The pups belonging to cohort 1 were euthanized on P7, immediately following the conclusion of the 48-h oxygen exposure and at least 4 h after their final drug or vehicle application, to collect samples for subsequent molecular biological analyses (*n* = 12/group). The pups in cohort 2 underwent the same regimen of oxygen exposure and drug/vehicle application as cohort 1; however, following the P7 treatments, they were allowed to recover in room air without any further manipulation or treatment until P14, when they were euthanized. Samples from cohort 2 were collected for both molecular biological analyses (*n* = 12/group) and immunohistochemical analyses (*n* = 8/group).

### 4.3. Drug Formulation and Dosing

Cannabidiol (CBD, Biomol GmbH, Hamburg, Germany, Cat. Cay90080) or the vehicle solution was administered at two distinct doses: 10 mg/kg body weight (BW), which corresponds to 0.1 mg/10 g BW per mouse, and 30 mg/kg BW, which corresponds to 0.3 mg/10 g BW per mouse. The maximum volume administered for any application was capped at 100 µL/10 g BW. The CBD was prepared by dissolving it in a specific solvent mixture composed of ethanol–Cremophor–PBS in a ratio of 1:1:18, utilizing Cremophor^®^ EL (Merck Millipore, Cat. 238470, Darmstadt, Germany). This identical solvent mixture, administered at a volume of 100 µL/10 g BW, constituted the vehicle that was applied to the control groups.

### 4.4. Sampling and Processing Heart Tissue

All pups were first brought into a state of deep anesthesia using a combination of ketamine (100 mg/kg) and xylazine (20 mg/kg) before undergoing transcardial perfusion. The pups designated for molecular biological analyses were perfused exclusively with cold phosphate-buffered saline (PBS, pH 7.4). The harvested tissue samples were immediately snap-frozen and stored at −80 °C until further analysis. In contrast, the pups intended for immunohistological analyses were additionally perfused with cold 4% (*w*/*v*) paraformaldehyde (PFA dissolved in PBS, pH 7.4) following the PBS washout to fix the tissues. Following harvesting, the hearts were subjected to a one-day post-fixation in PFA at 4 °C before being dehydrated in graded ethanol solutions and subsequently embedded in paraffin for histological processing.

### 4.5. RNA Extraction and qPCR

Heart tissue procurement for transcript analysis, including the prerequisite RNA isolation and subsequent reverse transcription into cDNA, was performed following methods previously established by our research group [31]. Briefly, total RNA was extracted from frozen tissue samples via acid phenol/chloroform extraction utilizing the RNA Solv Reagent (Omega Bio-Tek, Norcross, GA, USA, Cat. R6830). Two micrograms of the isolated, DNase-treated RNA were then reverse-transcribed, and the resulting cDNA was amplified and quantified using real-time quantitative polymerase chain reaction (qPCR). Amplification of the target gene sequences, which are detailed along with their abbreviations in Appendix A, was executed using the qPCRBIO SyGreen Mix Lo-ROX (PCR Biosystems, London, UK, Cat. PB20.11) on a QuantStudio™ 5 Real-Time PCR System (Applied Biosystems, Thermo Fisher Scientific Inc., Waltham, MA, USA). Gene expression analysis was conducted utilizing the 2^−ΔΔCT^ method [88], with standardized *18SrRNA* serving as the internal reference gene.

### 4.6. Sirius Red and HE Staining

Paraffin-embedded heart tissue blocks were sectioned at 5 µm thickness, subsequently deparaffinized, and then hydrated with descending alcohol concentrations (EtOH; 96%, 80%, 70%). After the sections were rinsed in distilled water, they were incubated with a saturated aqueous picric acid solution for at least 1 h. The picro-Sirius Red staining solution was prepared by dissolving 0.1% (*w*/*v*) of Sirius Red (Direct Red 80; Sigma-Aldrich, Taufkirchen, Germany, Cat. 365548) in saturated aqueous picric acid solution (approx. 1.3% in dH_2_O; Sigma-Aldrich, Cat# P6744), followed by filtration. The slides were then washed twice with freshly prepared acidified water (0.5% (*v*/*v*)) glacial acetic acid in ddH_2_O, hydrated again with an ascending concentration of ethanol (70%, 80%, 96%), and cleared with xylene substitute (ROTI^®^Histol, Carl Roth, Karlsruhe, Germany, Cat. 6640) before mounting on the slide with mounting medium.

Prior to HE staining, 5 µm thick paraffin-embedded heart tissue sections were deparaffinized and rehydrated. Deparaffinization was performed by immersing the sections twice in xylene substitute for 8 min each. Rehydration followed a graded ethanol series (all steps 3 min each): twice in 99% ethanol, twice in 96% ethanol, and sequentially in 70% and 50% ethanol. The sections were then rinsed with deionized water and stained with Mayer’s Hematoxylin (Sigma-Aldrich, Cat. 1.09249) for 8 min. Excess stain was removed by rinsing the slides under lukewarm running tap water for 8 min. Counterstaining involved immersion in aqueous Eosin (Morphisto GmbH, Offenbach am Main, Germany, Cat. 10177) containing 0.8% (*v*/*v*) glacial acetic acid for 1 min. The slides were then rinsed again with dH_2_O and quickly dehydrated in 70% ethanol (5 s), twice in 96% ethanol (5 s each), and twice in 99% ethanol (20 s each). Clearing was achieved by immersing the sections twice in xylene substitute for 2 min each. Finally, the sections were mounted using a resin-based mounting medium. All steps, unless otherwise specified, were performed at room temperature.

### 4.7. Aspects of Morphometric and Histological Analysis

Fibrosis quantification, specifically the red-stained collagen in the paraffin-embedded mouse heart sections, was conducted utilizing Sirius Red staining and ImageJ software (version 1.53 [89]). For each animal, the complete heart was captured for subsequent fibrosis analysis. Images were acquired at 10× magnification in the light field, employing both stitching and Z-stack techniques (BZX800 microscope, Keyence, Osaka, Japan). To accurately determine the fibrotic percentage per area, a calibration scale was initially established within ImageJ by measuring and saving an object of known dimensions as a reference standard. The Sirius Red-positive areas were isolated by applying the “colour convolution2” tool to set the threshold value. Data acquisition was performed in a standardized manner, recording both the percentage of the fibrotic area within the image frame and the area of the corresponding heart section. This methodology allowed for the final quantification of the fibrotic area expressed as a percentage per area [mm^2^]. All subsequent measurements were performed under blinded conditions. Mean values for each biological sample were computed by averaging the measurements obtained from all analyzed sections originating from the same animal. These mean values were then used to compare the data between the treated animals and their respective controls. The average fibrosis percentage of the control group was designated as the 100% baseline for normalization.

Left ventricular wall thickness was determined using the HE-stained heart sections (5 µm), visualized via light microscopy, and analyzed using ImageJ software. For each animal, transversal (short-axis) heart sections were used, ensuring the section was representative of the maximal ventricular dimension. Images of the entire section were acquired at 100× magnification in the light field, employing both stitching and Z-stack techniques (BZX800 microscope). To accurately convert pixel data into metric units, the system was calibrated by measuring and saving an object of known dimensions as a reference standard in ImageJ. The left ventricular wall thickness was measured as the perpendicular distance (orthogonale) between the endocardial surface and the epicardial surface of the free left ventricular wall. To ensure regional consistency, measurements were taken at four defined points along the wall for each section. A total of three serial sections per animal were analyzed, resulting in 12 individual measurement data points per animal. All measurements were performed under blinded conditions. The mean thickness for each animal was computed by averaging the values obtained from all measurements across the three analyzed sections. These mean values were then used for the comparison between the treated and control animals. For normalization, the average left ventricular wall thickness of the control group was designated as the 100% baseline.

### 4.8. Immunohistochemistry, Image Acquisition, and Quantification

Paraffin-embedded heart tissue was serially sectioned at 5 µm thickness and mounted onto SuperFrost Plus slides (Menzel, Braunschweig, Germany). The sections were deparaffinized twice in ROTI^®^Histol (Carl Roth) for 10 min each, followed by rehydration through a graded ethanol series (100%, 100%, 90%, 80%, 70%) for 3 min per step. For antigen retrieval, the sections were fixed in citrate buffer (pH 6.0) using a microwave oven for 10 min at 600 W. The slides were cooled for 30 min at RT and subsequently washed three times with PBS. Non-specific binding was minimized by blocking the sections with blocking buffer (10% (*v*/*v*) goat serum, 1% (*w*/*v*) bovine serum albumin, and 0.3% (*v*/*v*) Triton X-100 in PBS) for 1 h at RT. The sections were incubated overnight at 4 °C with the primary antibody, monoclonal rabbit anti-mouse Ki67 (1:250, Cell Signal Technology, Danvers, MA, USA, Cat. 9129S), diluted in antibody diluent (Zymed Laboratories, San Francisco, CA, USA, Cat. ZUC103). Following three washes in PBS, the sections were incubated with the secondary antibody, Alexa Fluor 594-conjugated goat anti-rabbit IgG (1:200; Thermo Fisher Scientific, Schwerte, Germany, Cat. A11037), diluted in antibody diluent for 1 h at RT. The nuclei were counterstained by incubating the sections with 4′,6-diamidino-2-phenylindole (DAPI, 1:1000; Sigma-Aldrich, Cat. D9542) for 10 min at RT. After three final PBS washes, the sections were mounted. A negative control, with the primary antibody omitted, confirmed the absence of non-specific immunoreactivity.

The heart slides were analyzed under blinded conditions using a Keyence fluorescent microscope (BX800) with BZ-II Viewer (version 01.03.00.01) and BZ-II Analyzer (version 1.1.2.4) software. Images were acquired using a 10× objective lens (100× magnification). For each transversal heart section, the entire area was captured via automatic stitching and saved as individual RGB files using identical exposure and display parameters. To quantify Ki67-positive proliferating cells, three separate sections of the whole heart per animal were analyzed. The DAPI signal was used to define cell nuclei. The number of Ki67-positive cells was quantified for each section using ImageJ (after minimal contrast adjustment) and expressed relative to the actual tissue area. The mean value per animal was calculated by averaging the counts across all analyzed sections, which was used for subsequent comparison between the treated and control groups.

### 4.9. Statistical Analysis

Prior to the commencement of this study and application for approval, the necessary sample size for each experimental group was calculated using G*Power V3.1.2, and this determined group size was strictly maintained. For data visualization and statistical evaluation, box and whisker plots were utilized. The box within these plots represents the interquartile range (IQR), with the horizontal line indicating the median. The whiskers depict the overall data spread beyond the first and third quartiles. Group comparisons were conducted using a one-way analysis of variance (ANOVA). In instances where the data showed a non-Gaussian distribution, the Kruskal–Wallis test was employed. Alternatively, if heterogeneity of variances was detected between groups, the Brown–Forsythe test was used. The subsequent pairwise multiple comparisons were selected according to the initial ANOVA result, utilizing either the Bonferroni, Dunn’s, or Dunnett T3 post hoc test. A probability value (*p*-value) less than 0.05 (*p* < 0.05) was established as the criterion for statistical significance. All figures and statistical calculations were generated using GraphPad Prism software (version 8.0, GraphPad Software, La Jolla, CA, USA).

## 5. Conclusions

In summary, this study demonstrates that neonatal hyperoxia induces persistent oxidative, inflammatory, and structural cardiac damage and that CBD administration attenuates these effects in a dose-dependent manner, preserves cardiomyocyte proliferation, and reduces maladaptive remodeling at optimal dosages. These findings underscore the potential of CBD as a cardioprotective intervention in the neonatal period, provided that dosage, timing, and developmental safety are carefully considered. Translation into clinical practice will require rigorous functional, mechanistic, and safety studies. Nonetheless, the present data support further investigation of CBD in the context of neonatal oxygen therapy and cardiac protection.

## Figures and Tables

**Figure 1 ijms-27-00146-f001:**
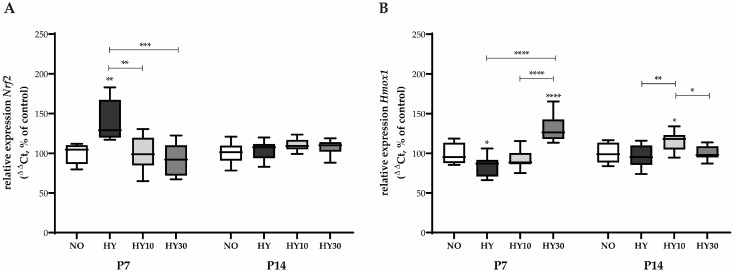
Quantification of oxidative stress regulating transcripts for (**A**) *Nrf2* and (**B**) *Hmox1* of whole hearts for P7 after 2 days of oxygen exposure and recovery until P14 using qPCR. Data are normalized to the level of mouse pups exposed to normoxia at each time point (control 100%, NO, 21% O_2_, white bars) with verum groups hyperoxia (HY, 80% O_2_, black), hyperoxia with CBD 10 mg/kg (HY10, light grey), and hyperoxia with CBD 30 mg/kg (HY30, dark grey). Data are presented as box–whisker plots, with *n* = 12 per group. * *p* < 0.05, ** *p* < 0.01, *** *p* < 0.001, and **** *p* < 0.0001 (ANOVA; Kruskal–Wallis).

**Figure 2 ijms-27-00146-f002:**
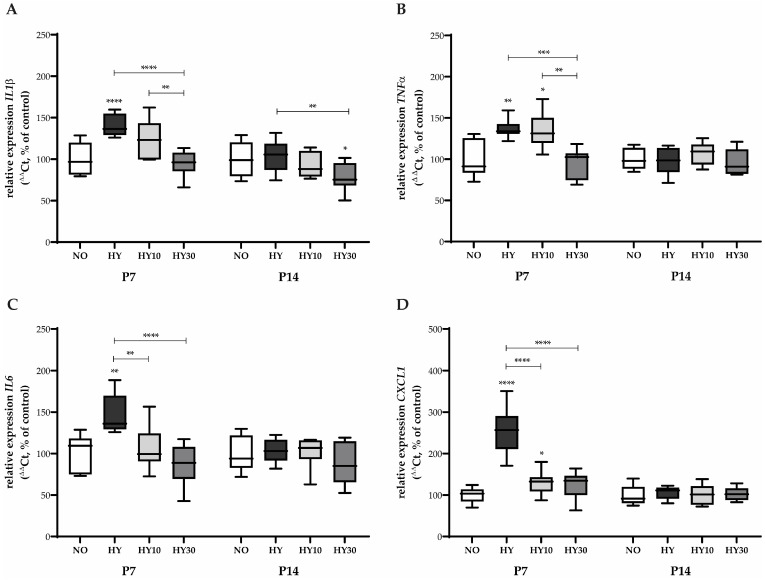
Quantification of inflammatory-regulating transcripts for (**A**) *IL1β*, (**B**) *TNFα*, (**C**) *IL6*, and (**D**) *CXCL1* of whole hearts for P7 after 2 days of oxygen exposure and recovery until P14 using qPCR. Data are normalized to the level of mouse pups exposed to normoxia at each time point (control 100%, NO, 21% O_2_, white bars) with verum groups hyperoxia (HY, 80% O_2_, black), hyperoxia with CBD 10 mg/kg (HY10, light grey), and hyperoxia with CBD 30 mg/kg (HY30, dark grey). Data are presented as box–whisker plots, with *n* = 12 per group. * *p* < 0.05, ** *p* < 0.01, *** *p* < 0.001, and **** *p* < 0.0001 (ANOVA; Kruskal–Wallis; Brown–Forsythe).

**Figure 3 ijms-27-00146-f003:**
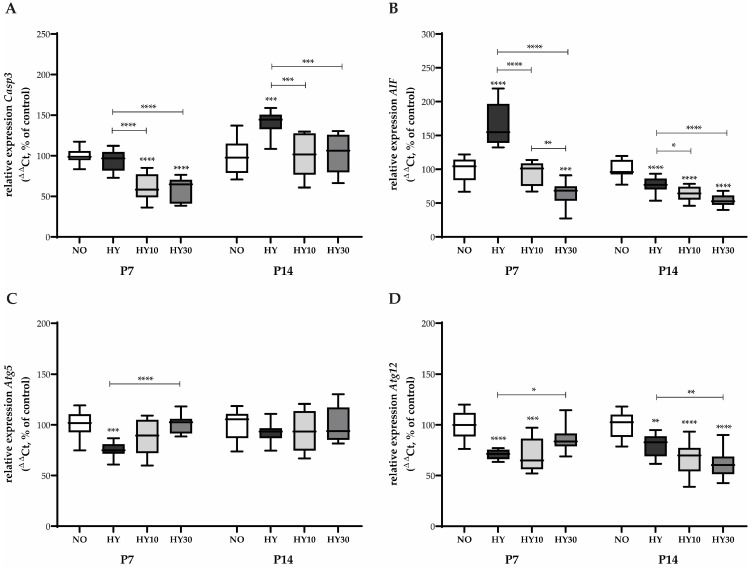
Quantification of cell death regulating transcripts for (**A**) *Casp3*, (**B**) *AIF*, (**C**) *Atg5*, and (**D**) *Atg12* of whole hearts for P7 after 2 days of oxygen exposure and recovery until P14 using qPCR. Data are normalized to the level of mouse pups exposed to normoxia at each time point (control 100%, NO, 21% O_2_, white bars) with verum groups hyperoxia (HY, 80% O_2_, black), hyperoxia with CBD 10 mg/kg (HY10, light grey), and hyperoxia with CBD 30 mg/kg (HY30, dark grey). Data are presented as box–whisker plots, with *n* = 12 per group. * *p* < 0.05, ** *p* < 0.01, *** *p* < 0.001, and **** *p* < 0.0001 (ANOVA; Brown–Forsythe).

**Figure 4 ijms-27-00146-f004:**
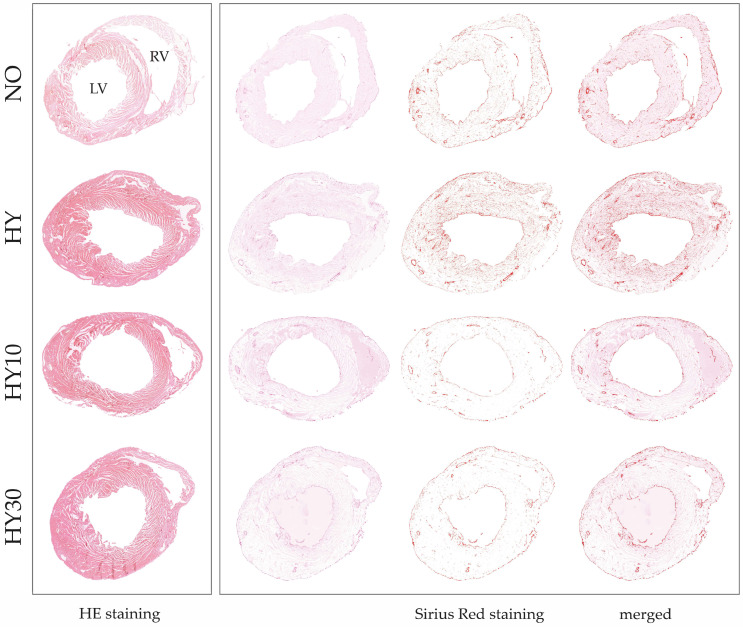
Representative cross-sections of the neonatal myocardium on postnatal day 14 (P14), following a 48-h hyperoxia (HY) exposure period (P5–P7) and a recovery phase, shown with normoxia (NO, 21% O_2_), with verum groups hyperoxia (HY, 80% O_2_), hyperoxia with CBD 10 mg/kg (HY10), and hyperoxia with CBD 30 mg/kg (HY30). HE staining (left panels) provides an overview of tissue architecture, cellular morphology, and chamber geometry, while Sirius Red staining (right panels) specifically visualizes collagen deposition, serving as a marker for myocardial fibrosis. All cardiac cross-sections were obtained at the midventricular level and are presented in a standardized orientation. LV, left ventricle; RV, right ventricle; 100× magnification.

**Figure 5 ijms-27-00146-f005:**
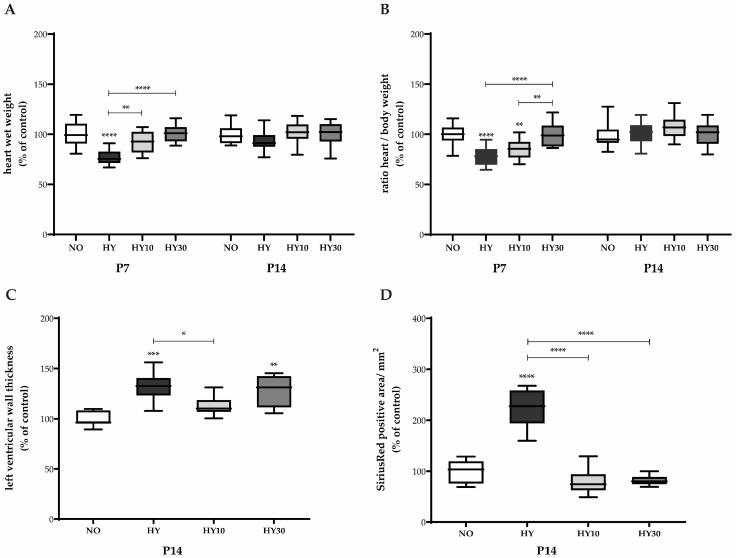
Quantification of (**A**) heart wet weight, (**B**) ratio heart/body weight, (**C**) left ventricular wall thickness, and (**D**) Sirius Red-positive area of hearts for P7 after 2 days of oxygen exposure and/or recovery until P14. Data are normalized to the level of mouse pups exposed to normoxia at each time point (control 100%, NO, 21% O_2_, white bars) with verum groups hyperoxia (HY, 80% O_2_, black), hyperoxia with CBD 10 mg/kg (HY10, light grey), and hyperoxia with CBD 30 mg/kg (HY30, dark grey). Data are presented as box–whisker plots, with *n* = 12 per group. * *p* < 0.05, ** *p* < 0.01, *** *p* < 0.001, and **** *p* < 0.0001 (ANOVA; Kruskal–Wallis).

**Figure 6 ijms-27-00146-f006:**
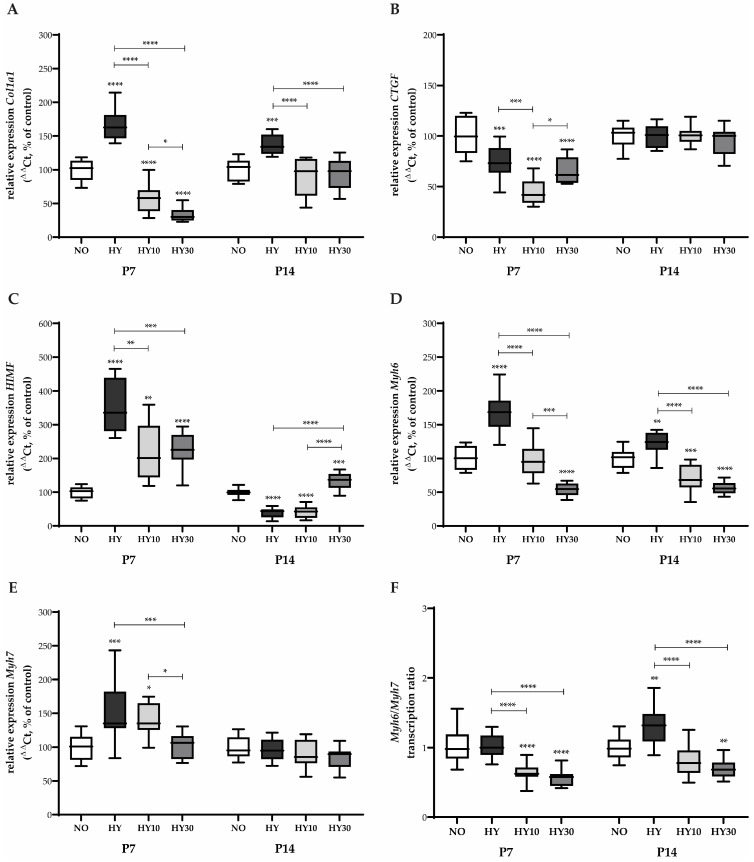
Quantification of cardiac remodeling transcripts for (**A**) *Col1a1*, (**B**) *CTGF*, (**C**) *HIMF*, (**D**) *Myh6*, (**E**) *Myh7*, and (**F**) *Myh6/Myh7* ratio of whole hearts for P7 after 2 days of oxygen exposure and recovery until P14 using qPCR. Data are normalized to the level of mouse pups exposed to normoxia at each time point (control 100%, NO, 21% O_2_, white bars) with verum groups hyperoxia (HY, 80% O_2_, black), hyperoxia with CBD 10 mg/kg (HY10, light grey), and hyperoxia with CBD 30 mg/kg (HY30, dark grey). Data are presented as box–whisker plots, with *n* = 12 per group. * *p* < 0.05, ** *p* < 0.01, *** *p* < 0.001, and **** *p* < 0.0001 (ANOVA; Brown–Forsythe).

**Figure 7 ijms-27-00146-f007:**
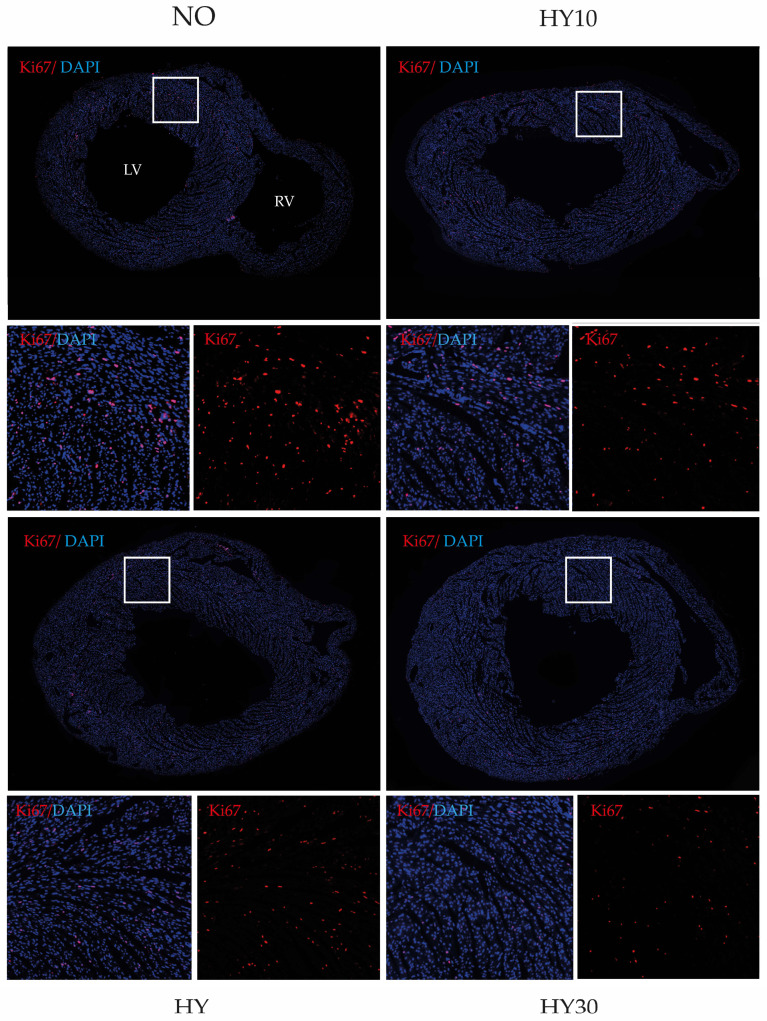
Representative cross-sections of the neonatal myocardium on postnatal day 14 (P14), following a 48-h hyperoxia (HY) exposure period (P5–P7) and a recovery phase labeled with nuclear-localized proliferation marker Ki67 (red) and DAPI (blue), shown with normoxia (NO, 21% O_2_, upper left), with verum groups hyperoxia (HY, 80% O_2_, lower left), hyperoxia with CBD 10 mg/kg (HY10, upper right), and hyperoxia with CBD 30 mg/kg (HY30, lower right). The outlined regions (white bordered boxes) are magnified 4.3× compared to the original image. All cardiac cross-sections were obtained at the midventricular level and are presented in a standardized orientation. LV, left ventricle; RV, right ventricle; 100× magnification.

**Figure 8 ijms-27-00146-f008:**
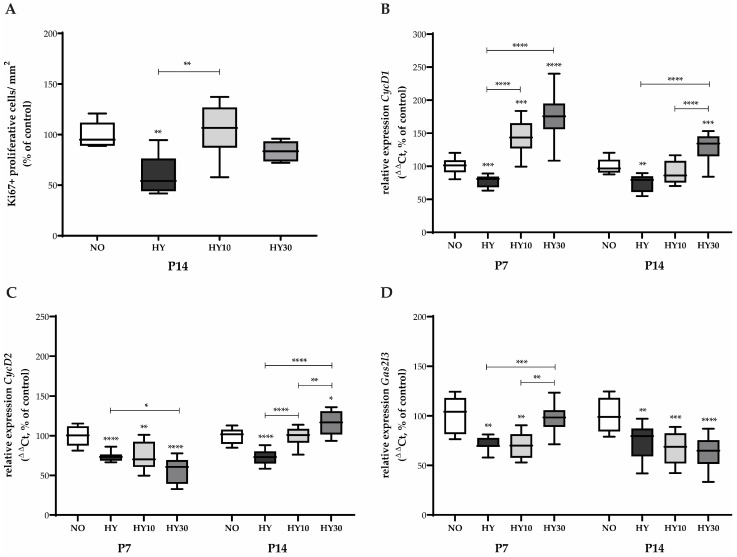
Quantification of (**A**) Ki67 immunostaining and cell cycling transcripts for (**B**) *CycD1*, (**C**) *CycD2*, and (**D**) *Gas2l3* of whole hearts for P7 after 2 days of oxygen exposure and recovery until P14 using qPCR. Data are normalized to the level of mouse pups exposed to normoxia at each time point (control 100%, NO, 21% O_2_, white bars) with verum groups hyperoxia (HY, 80% O_2_, black), hyperoxia with CBD 10 mg/kg (HY10, light grey), and hyperoxia with CBD 30 mg/kg (HY30, dark grey). Data are presented as box–whisker plots, with *n* = 12 per group. * *p* < 0.05, ** *p* < 0.01, *** *p* < 0.001, and **** *p* < 0.0001 (ANOVA; Brown–Forsythe).

**Figure 9 ijms-27-00146-f009:**
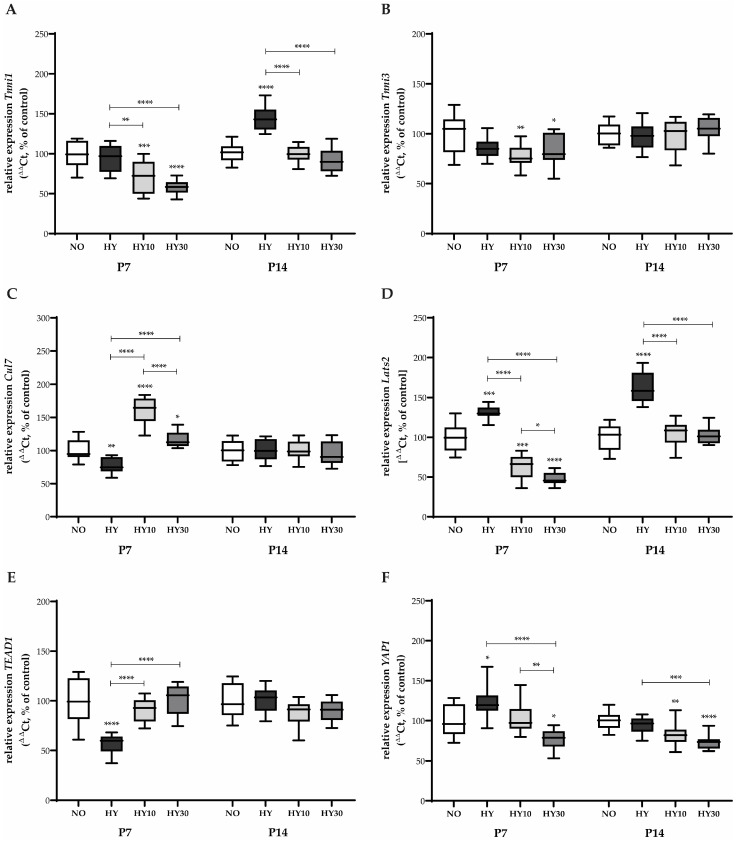
Quantification of cardiac functioning transcripts for (**A**) *Tnni1*, (**B**) *Tnni3*, (**C**) *Cul7*, (**D**) *Lats2*, (**E**) *TEAD1*, and (**F**) *YAP1* of whole hearts for P7 after 2 days of oxygen exposure and recovery until P14 using qPCR. Data are normalized to the level of mouse pups exposed to normoxia at each time point (control 100%, NO, 21% O_2_, white bars) with verum groups hyperoxia (HY, 80% O_2_, black), hyperoxia with CBD 10 mg/kg (HY10, light grey), and hyperoxia with CBD 30 mg/kg (HY30, dark grey). Data are presented as box–whisker plots, with *n* = 12 per group. * *p* < 0.05, ** *p* < 0.01, *** *p* < 0.001, and **** *p* < 0.0001 (ANOVA; Brown–Forsythe).

## Data Availability

The original contributions presented in this study are included in the article/Appendix A. Further inquiries can be directed to the corresponding author.

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
