# Peer review of "Cannabidiol Protects the Neonatal Mouse Heart from Hyperoxia-Induced Injury"

_ijms, 2025, doi:10.3390/ijms27010146_

Round 1
Reviewer 1 Report
Comments and Suggestions for Authors
Dear authors of work ijms-4040864, ​​I have some constructive feedback on your work:
Lines 46-50: The idea is not clear. I suggest better connecting it to the previous idea.
Lines 51-54: The statements require bibliographic citations.
Lines 58-60: The statements require bibliographic citations.
Lines 60-61: The statements require bibliographic citations.
Lines 99-101: The statements require bibliographic citations.
Lines 103-104: The statements require bibliographic citations.
Figure 4: I suggest adding a compass and describing the anatomical structures (in the control).
Figure 7: I suggest adding a compass and a dimensional bar. It would be desirable to show higher magnifications (400x). Nuclear immunodetection should be clearly shown.
Figure 8: I suggest moving Figure 8A to Figure 7.
Reviewer 2 Report
Comments and Suggestions for Authors
Article is overall well-written, with interesting innovative topic attractive for readers. Methodologically well-designed with minor issues that should be addressed:
- Abstract section is rather descriptive, please include some main findings in terms of parameters that were changed and p-values.
- In order to investigate the dose-dpeendent effect of CBD the authors should have test three doses. Please, explain the rationale for using only two doses.
- The following sentence `Prior to the commencement of the study and application for approval, the necessary sample size for each experimental group was calculated using G*Power V3.1.2, and this determined group size was strictly maintained.` should be transferred into subsection Statistics.
- Were the animals aclimatized prior to inclusion to study? Specify the food they used in terms of nutrients percentage.
- Section 2.9. belongs to Discussion section as it describes main findings, not in the Results section. Please, correct.
- No hemodynamic assesment was done, which would give greater picture about cardioprotective effect of CBD.
- Authors investigasted antifibrotic effect of CBD also. If possible include TGF beta measurement.
- Discussion is overall well-written, in details with mechanistic approach.
Satisfactory
Reviewer 3 Report
Comments and Suggestions for Authors
Comments:
In the article titled "Cannabidiol Protects the Neonatal Mouse Heart from Hyperoxia-Induced Injury," the author has effectively presented data that aligns with the main discoveries. Notably, Cannabidiol (CBD) is a non-psychoactive phytocannabinoid known for its antioxidant and anti-inflammatory properties, yet it has seldom shown protective effects against hyperoxic damage to the developing heart in neonates, making this article particularly significant for readers. This article is worth of publication as it excels in all journal criteria. The results indicates that CBD may offer cardioprotective benefits in cases of oxidative and inflammatory heart injury in newborns, although its effectiveness is influenced by both dosage and developmental stage. Few of the key finding which is greater impact are:
1) CBD enhanced antioxidant defenses through Nrf2/ARE signaling and decreases ROS/RNS formation in neuronal and cardiovascular models. 2) Higher doses of CBD treatment reduced the levels of IL1β, TNFα, and CXCL1 cytokines, showcasing its anti-inflammatory properties. 3) CBD also bolstered cell survival signals: the autophagy markers Atg5/12, suppressed by hyperoxia, were restored by CBD, and the caspase-independent apoptosis mediator AIF was reduced. 4) At a dosage of 10 mg/kg, CBD was able to prevent hypertrophy and fibrosis while maintaining proliferative capacity. 5) Female neonatal cardiomyocytes displayed lower apoptosis rates and better recovery after oxidative stress.
While the constraints on drug dosage variation and timing may pose a limitation, these findings can serve as a foundation for future research.
Round 2
Reviewer 2 Report
Comments and Suggestions for Authors
Authors have succesfully endorsed the majority of reviewer`s comments, so I consider that article is much improved and now suitable for publication.